# Differences in Kinanthropometric Variables and Physical Fitness of Adolescents with Different Adherence to the Mediterranean Diet and Weight Status: “Fat but Healthy Diet” Paradigm

**DOI:** 10.3390/nu15051152

**Published:** 2023-02-24

**Authors:** Adrián Mateo-Orcajada, Raquel Vaquero-Cristóbal, Jesús Miguel Montoya-Lozano, Lucía Abenza-Cano

**Affiliations:** 1Facultad de Deporte, UCAM Universidad Católica de Murcia, 30107 Murcia, Spain; 2IES Arzobispo Lozano, 30520 Murcia, Spain

**Keywords:** body weight, dietary pattern, gender differences, physical activity, physical condition, youth

## Abstract

The present investigation provides a new paradigm, the fat but healthy diet, through which to analyze the importance of adherence to the Mediterranean diet (AMD) in the adolescent population. To this end, the objectives were to analyze the existing differences in physical fitness, level of physical activity, and kinanthropometric variables in males and females with different AMD and to determine the existing differences in physical fitness, level of physical activity, and kinanthropometric variables in adolescents with different body mass index and AMD. The sample consisted of 791 adolescent males and females whose AMD, level of physical activity, kinanthropometric variables, and physical condition were measured. The results showed that when analyzing the whole sample, the differences were only significant in the level of physical activity among adolescents with different AMD. However, when considering the gender of the adolescents, the males also showed differences in the kinanthropometric variables, while the females did so in the fitness variables. In addition, when considering gender and body mass index, the results showed that overweight males with better AMD showed less physical activity and higher body mass, sum of three skinfolds, and waist circumference, and females did not show differences in any variable. Therefore, the benefits of AMD in anthropometric variables and physical fitness of adolescents are questioned, and the fat but healthy diet paradigm cannot be confirmed in the present research.

## 1. Introduction

In the adolescent population, nutritional habits are one of the most important factors for the establishment of healthy lifestyles [1,2]. Thus, a correct diet facilitates the prevention of chronic diseases such as obesity [3], contributes to better glycemic control [4], and has a fundamental anti-inflammatory and antioxidant effect in this population [5].

In recent decades, the adherence to the Mediterranean diet (AMD) of adolescents has been used in Europe as a criterion for assessing their diet because it is one of the healthiest dietary patterns known to date [6,7]. Previous research conducted in adolescents has tried to establish differences in AMD according to gender [8,9] and to analyze the relationship between AMD and other determinant variables for health such as body composition [10,11], level of physical activity [11,12], or physical fitness [13].

Regarding some components of physical fitness and physical activity levels of adolescents according to AMD, the results found were very disparate. Some of the results found in previous research were: (1) higher values in handgrip strength and vertical jump in males with a higher AMD but not in females [13]; (2) a higher performance in cardiorespiratory endurance tests in males and females with moderate-high AMD [10,13]; (3) a higher level of physical activity in adolescents with a higher AMD [12]; or (4) absence of significant differences in the level of physical activity and physical fitness among adolescents with different AMD [11].

Similarly, the existing relationship between body composition and AMD shows contradictory results. Some previous research found: (1) no significant differences in males or females in body composition when considering the level of AMD [10,11]; and (2) significant differences in fat percentage when considering AMD, with males with moderate-high AMD showing the lowest fat percentage [13,14].

Although previous research has investigated the differences between males and females in AMD as well as the differences in body composition, level of physical activity, and physical fitness of adolescents according to AMD, the conclusions are not clear. In the field of physical activity, a paradigm known as “fat but fit” has been considered in recent years [15,16]. In this paradigm, overweight and obese adolescents with a better level of physical fitness showed lower cardiometabolic risks than adolescents with the same weight status but with a worse level of physical fitness [15,16]. Extrapolating this theory to the field of nutrition, it could be that a similar phenomenon occurs, whereby differences in body composition, physical fitness, and physical activity level of adolescents could differ according to AMD and adolescent weight. Thus, a paradigm called “fat but healthy diet” could be proposed in which, hypothetically, adolescents with optimal AMD would present higher levels of physical activity, better kinanthropometric variables, and higher performance in fitness tests compared to adolescents with worse AMD within the same weight status group. It could provide key information on the relevance of diet in the adolescent population regardless of weight status. However, no known study has addressed this joint approach to AMD and weight status.

For this reason, the main objectives of the present investigation were (a) to analyze the existing differences in physical fitness, level of physical activity, and kinanthropometric variables in males and females with different AMD and (b) to determine the existing differences in physical fitness, level of physical activity, and kinanthropometric variables in adolescents with different body mass index (BMI) and AMD.

Based on previous research, the following research hypotheses are posed: (H1) differences will be significant in physical fitness, physical activity level, and fat-mass-related variables in adolescents according to AMD level, although there will be differences according to gender; and (H2) adolescents with higher AMD will perform more physical activity, present better body composition, and higher performance in physical fitness tests regardless of their weight status.

## 2. Materials and Methods

### 2.1. Design

A descriptive and cross-sectional design with non-probabilistic convenience sampling was carried out. Before starting the study, the institutional ethics committee of the Catholic University of Murcia approved the protocol designed for the research study (protocol code: CE022102, 26 February 2021) according to the Helsinki declaration. The STROBE statement was followed for the development of the manuscript [17].

### 2.2. Participants

The minimum sample size was calculated using the statistical software Rstudio 3.15.0 (Rstudio Inc., Boston, MA, USA) using the standard deviations from previous studies that analyzed diet in adolescents (SD = 2.32) [10]. The minimum sample size for the development of the present research was 750 adolescents considering an estimated error (d) of 0.22 for a 99% confidence interval.

The final sample consisted of 791 adolescents (404 males and 387 females; mean age: 14.39 ± 1.26 years) who decided to voluntarily participate in the study. Informed consent was signed before the start of the study by the adolescents and their parents, accrediting their participation in the study. The participants were enrolled in four secondary schools in the (Region of Murcia) (two located in the north, one in the center, and one in the southeast). These schools were selected because they had a high number of students enrolled in compulsory secondary education. In Spain, during this formative stage, students receive notions of nutrition and dietetics in the subjects of Biology and Geology as well as in Physical Education [18]. Adolescents learn about the importance of macronutrients and micronutrients of general importance for the functionality of the body as well as the importance of a healthy diet in lifestyle habits and health improvement, but this is always learned in a secondary and complementary way and not as the main content of any of the aforementioned subjects.

The inclusion criteria of the sample were as follows: (a) enrolled in compulsory secondary education with ages between twelve and sixteen years old and (b) not having any incapacitating disease that would make it impossible to complete the questionnaires and physical tests.

### 2.3. Instruments

#### 2.3.1. Questionnaire Measurements

The KIDMED questionnaire [6] was used to determine the AMD of these adolescents. This questionnaire presents moderate reliability and reproducibility values for use in adolescents (α = 0.79 and kappa: 0.66). The questionnaire is composed of 16 questions that were rated by the adolescents with a score of 1 or 0 depending on whether the criterion was met. Subsequently, the scores obtained were added up considering that twelve of the questions had a positive connotation (+1) (favoring a good adherence), and four had a negative connotation (−1) (favoring an inadequate adherence). The final score was between 0 and 12 points for all the participants, establishing three classifications: poor adherence to the Mediterranean diet (0–3 points), need to improve adherence (4–7 points) or optimal adherence (8–12 points) [6].

The level of physical activity was determined using the Spanish version of the “Physical Activity Questionnaire for Adolescents” (PAQ-A) [19]. This questionnaire has an intraclass correlation coefficient of 0.71 and an internal consistency of 0.74 for the final score. It is composed of nine questions that provide information on the physical activity performed in the last seven days, considering different time slots during the day. A Likert scale of 1 to 5 points is used for its completion, with 1 being an absence of physical activity and 5 a high level of physical activity [19].

#### 2.3.2. Kinanthropometric Measurements

The kinanthropometric analysis was composed of (1) three basic measurements (body mass, height, and sitting height); (2) three skinfolds (triceps, thigh, and calf); and (3) five girths (arm relaxed, waist, hip, thigh, and calf). To carry them out, the protocol established by the International Society for the Advancement of Kinanthropometry (ISAK) was followed [20]. ISAK-accredited anthropometrists (levels 2 to 4) measured each variable twice, performing a third measurement when the differences between the first and second measurements were greater than 5% in the skinfolds and 1% in the rest of the measurements. The mean of the measured values was used when two measurements were performed, while the median was used when a third measurement was performed [20].

The variables from the measurements were used to calculate BMI, Σ3 skinfolds (triceps, thigh, and calf), corrected girths of the arm [arm relaxed girth − (π × triceps skinfold)], thigh [middle thigh girth − (π × thigh skinfold)], and calf [calf girth − (π × calf skinfold)], fat mass (%) [21], muscle mass [22], and waist-to-hip ratio (waist girth/hip girth).

The intra- and inter-evaluator technical error of measurements (TEM) were calculated in a sub-sample. The intra-evaluator TEM was 0.02% for the basic measurements; 1.21% for skinfolds, and 0.04% for the girths; and the inter-evaluator TEM was 0.03% for the basic measurements; 1.98% for skinfolds, and 0.06% for the girths.

The kinanthropometric equipment used was calibrated prior to the measurements. A TANITA BC 418-MA Segmental (TANITA, Tokyo, Japan), with an accuracy of 100 g, was used for body mass. For height and sitting height, a SECA stadiometer 213 (SECA, Hamburg, Germany) with an accuracy of 0.1 cm was used. A skinfold caliper (Harpenden, Burgess Hill, UK) was used for measuring skinfolds, with an accuracy of 0.2 mm. An inextensible tape (Lufkin W606PM, Missouri City, TX, USA) was used to measure girths with a 0.1 cm accuracy.

#### 2.3.3. Physical Fitness Test

The familiarization and correct performance of the physical fitness tests by the adolescents was supervised by four investigators with previous experience in the field. Each investigator oversaw the same physical fitness tests during all measurements to avoid inter-evaluator error.

Three physical fitness tests were performed, which were chosen for their validity and reliability in this population [23,24,25]. The 20 m shuttle run test was chosen to assess cardiorespiratory endurance in adolescents. This is an incremental test that consists of running twenty meters as many times as possible. This test ends when the distance is not covered two consecutive times before the allotted time ends or when the adolescent reaches exhaustion. The formula by Leger et al. [26] was used to determine the maximum oxygen consumption (VO_2_ max) of each adolescent.

Handgrip strength was assessed using the handgrip strength test, in which the adolescents had to apply the greatest possible force on a Takei Tkk5401 digital handheld dynamometer (Takei Scientific Instruments, Tokyo, Japan). The adolescents’ elbow was fully extended, as this is the optimal position for applying the maximum force [27].

The countermovement jump (CMJ) was used to assess the explosive power of the lower limbs. For its execution, following the protocol by Barket et al. [28], the adolescents had to perform a maximal vertical jump. The adolescents’ hands were to be placed at the waist, and the legs and back must be fully extended during the flight phase. In the starting position, the adolescents had to stand on the force platform (MuscleLab, Stathelle, Norway) with hands on their waist and feet hip-width apart. Subsequently, they performed a knee flexion to 90° as quickly as possible, followed by a full knee extension to reach the maximum possible height in the vertical jump.

### 2.4. Procedure

The tests were carried out during Physical Education class time using covered sports pavilions of the participating compulsory secondary education centers to reduce contaminating variables that could affect the results.

First, all the adolescents completed the KIDMED and PAQ-A questionnaires. Subsequently, the kinanthropometric measurements were taken. Next, the correct execution of the handgrip strength and CMJ tests was explained to the students so that they became familiar with them. Once the familiarization process was completed, a warm-up consisting of running and joint mobility exercises was carried out, after which the tests were performed. Finally, the 20 m shuttle run test was performed. All the physical condition tests were performed twice, leaving two minutes of recovery time between the two measurements of each test and five minutes between the different tests. The best value of each test was recorded, except for the 20 m shuttle run test, which was performed only once. The testing protocol was established based on previous research [29] and following the recommendations of the National Strength and Conditioning Association (NSCA). These recommendations consider the fatigue generated by each test and establish sufficient recovery time between them to minimize possible interferences [30].

### 2.5. Data Analysis

The Kolmogorov–Smirnov test was used to assess the normality of the data. As all variables showed a normal distribution, parametric tests were used to analyze them. Descriptive statistics were used to find the mean and standard deviations. An ANOVA was performed to establish the existing differences in the physical activity level, physical condition, and kinanthropometric variables according to the AMD of adolescents. Next, an ANCOVA was performed to determine the existing differences in the measured variables as a function of AMD, considering gender and BMI as covariates of the model. Subsequently, a MANOVA was performed to determine the differences in the variables measured between males and females according to AMD and to establish the differences between the different weight statuses according to AMD in general and for males and females. The Bonferroni post hoc analysis was used to determine the differences between groups. Partial eta squared (η2) was used to establish whether the effect size was small (ES ≥ 0.10), moderate (ES ≥ 0.30), large (ES ≥ 1.2), or very large (ES ≥ 2.0), with an error of *p* < 0.05. A *p* < 0.05 value was used to determine statistical differences [31]. The SPSS statistical software was used to perform the statistical analysis (v.25.0; SPSS Inc., Chicago, IL, USA).

## 3. Results

### 3.1. Differences in the Study Variables According to the AMD Level

The differences in the level of physical activity, kinanthropometric variables, and physical fitness of adolescents with different levels of AMD are shown in Table 1. The differences were significant only in the level of physical activity, with the adolescents with an optimal adherence being those who practiced sports the most (*p* < 0.001). The inclusion of the covariates gender and BMI in the model showed significant differences for gender (*p* < 0.001–0.004) in all analyzed variables, except for BMI (*p* = 0.064) and hip girth (*p* = 0.121); however, when considering BMI, significant differences were found in all the variables (*p* < 0.001–0.013) except for height (*p* = 0.081).

Figure 1 shows the differences among males with poor AMD, males that need to improve AMD, and males with an optimal AMD as well as among females with poor AMD, females that need to improve their AMD, and females with an optimal AMD. With respect to males, the differences were significant in the level of physical activity and kinanthropometric variables but not in physical fitness. Females showed differences in the level of physical activity and physical fitness variables but not in kinanthropometric variables.

Bonferroni’s pairwise comparison showed that males with a poor AMD had a lower level of physical activity (*p* < 0.001–0.039), body mass (*p* = 0.032), BMI (*p* = 0.030), hip girth (*p* = 0.021), corrected thigh girth (*p* = 0.044), fat mass (*p* = 0.031), and muscle mass (*p* = 0.050) than males with an optimal and/or need to improve AMD. Regarding the females, whose who showed a poor AMD or need to improve AMD had a lower level of physical activity (*p* = 0.001–0.003) and VO_2_ max (*p* = 0.037) than females with an optimal AMD.

### 3.2. Differences in the Study Variables According to the Gender, AMD Level, and Weight Status

The differences in the analyzed variables according to the AMD and the BMI of the adolescents are shown in Figure 2. In the normal weight (*p* < 0.001) and underweight (*p* = 0.007–0.026) groups, adolescents with an optimal AMD showed a significantly higher level of physical activity. In the overweight group, adolescents with an optimal AMD showed significantly higher values in body mass (*p* = 0.014).

Table 2, Table 3 and Table 4 show the differences in the level of physical activity, kinanthropometric variables, and physical fitness in males and females who were normal weight, overweight and underweight with different levels of AMD. In normal weight males and females, differences were significant in the level of physical activity (*p* = 0.001–0.011), with males and females with an optimal AMD showing higher scores in both groups (Table 2). In the overweight group, significant differences were found in BMI (*p* = 0.027), sum of three skinfolds (*p* = 0.044), and waist girth (*p* = 0.016), with males with an optimal AMD showing higher values in all these variables (Table 3). In the underweight group, males with optimal AMD showed higher scores in the level of physical activity (*p* = 0.004), and males with a need to improve their AMD showed higher values in the CMJ test (*p* = 0.003) as compared to males with a poor AMD. Also in this group, males with an optimal AMD showed higher values of hip girth (*p* = 0.040) as compared to males with a need to improve AMD (Table 4). Females in the overweight and underweight groups did not present significant differences in any variable.

## 4. Discussion

The main objectives of the present investigation were (a) to analyze the existing differences in physical fitness, level of physical activity, and kinanthropometric variables in males and females with different AMD and (b) to determine the existing differences in physical fitness, level of physical activity, and kinanthropometric variables in adolescents with different BMI and AMD. Based on these objectives and on previous scientific literature, the following research hypotheses were established: (H1) differences will be significant in physical fitness, physical activity level, and fat-mass-related variables in adolescents according to AMD level, although there will be differences according to gender; and (H2) adolescents with higher AMD will perform more physical activity, present better body composition, and have higher performance in physical fitness tests regardless of their weight status.

According to the first objective of the present investigation (to analyze the existing differences in physical fitness, level of physical activity, and kinanthropometric variables in males and females with different AMD), the results showed only significant differences in the level of physical activity of the adolescents; adolescents with an optimal AMD practiced sports to a greater extent than those with a poor AMD. No differences were found in anthropometry and physical fitness variables. However, when considering the gender of the adolescents, both males and females with an optimal AMD presented a significantly higher level of physical activity. In addition, males with and optimal AMD showed greater muscle mass, especially in the thigh area, but also greater values in body mass, BMI, and fat mass, especially in the hip area, with respect to the poorer AMD group. Among females with optimal AMD, only differences were found in VO_2_ max with respect to the poorer AMD group. Previous research does not provide conclusive results in this regard, as some studies showed that there was no relationship between AMD, physical activity level, and kinanthropometric variables [11], while other studies showed that adolescents with better AMD performed more physical activity [12] and presented a higher VO_2_ max [10,13]. More specifically, the higher fat percentage of males with higher adherence to the Mediterranean diet may be because previous studies have suggested that high-fat diets such as the Mediterranean diet may promote obesity and fat accumulation when there is a positive energy balance [32]. Indeed, males with optimal AMD showed a greater increase in fat mass compared to muscle mass, which could be the origin of the changes in body mass and BMI [33]. In addition, this would explain why no differences were found in the fitness tests related to strength [34,35]. On the other hand, the isolated improvement in VO_2_ max in females could be due to the fact that adolescents with a better diet are those who are more aware of the importance of healthy habits [1,2], thus leading them to practice more physical activity. Therefore, it could be the greater practice of physical activity that is responsible for the higher VO_2_ max compared to the rest of the AMD groups [36,37]. Therefore, differences obtained in the present study could indicate that AMD as an isolated factor is not a determinant in the changes in kinanthropometric variables or in the fitness of adolescents. Despite these results, questions remain for future studies.

The results obtained in the present study partially confirm the first research hypothesis (H1) since adolescents with better AMD had a higher level of physical activity. However, the differences were not significant in the kinanthropometric or physical condition variables. When analyzing the results according to the gender of the adolescents, the differences were significant in males and females with optimal AMD compared to those with worse AMD. In this regard, males and females with optimal AMD showed a higher level of physical activity, but only males showed differences in body composition (increasing fat mass to a greater extent than muscle mass), and only females showed differences in physical fitness (increasing VO_2_ max) but not being able to claim that this was due to better AMD.

The second objective of the present study was to determine the existing differences in physical fitness, level of physical activity, and kinanthropometric variables of adolescents with different BMI and AMD, which could be termed the “fat but healthy diet” paradigm. Following the line of the “fat but fit” paradigm [15,16], it was hypothesized that adolescents with a better AMD would show a higher level of physical activity, better kinanthropometric variables, as well as a higher performance in the fitness variables compared to adolescents presenting worse AMD within the same weight status. Thus, the obtained results showed higher levels of physical activity in the optimal AMD group of normal weight and underweight adolescents. However, this was not the case in the overweight group, where a higher body mass was also found in the optimal AMD group. Previous research that considered adolescent AMD showed a higher level of physical activity in the group of adolescents with a higher AMD [12]. It is important to note that adolescent BMI and AMD have not been previously considered together, so the results obtained in this regard are novel. The fact that the overweight group was the only one that did not show significant differences in the level of physical activity among adolescents with different AMD could be explained by the frequent alterations in body image suffered by overweight and obese adolescents. This has very negative consequences during adolescence, mainly related to dietary alterations and avoidance of sports participation [38]. Regarding the greater body mass in adolescents with an optimal AMD in the overweight group, a possible explanation would be that the type of food ingested was not as decisive as the quantity ingested [39]. Thus, the lack of physical activity in this population, linked to excessive intake, would favor the increase in body mass, although future research that analyzes the specific daily intake of adolescents is necessary to corroborate this conclusion.

When analyzing the results considering the BMI and gender of the adolescents, it should be noted that the differences in physical fitness were significant in males and females, while kinanthropometric variables only showed differences in the group of males. Regarding physical fitness, the CMJ of underweight males with a poor AMD was significantly lower than the rest of the males with a better AMD. Previous research analyzing CMJ performance found no significant differences in either males or females based on AMD [13]. These results suggest that AMD may not be particularly relevant in this variable, so the observed differences could be due to the fact that males in the underweight group have less muscle mass and corrected thigh and calf girth than males in the normal weight and overweight groups. This could be a determining factor in the relationship between the amount of muscle mass and CMJ performance [40].

In the group of overweight males with an optimal AMD, regarding the kinanthropometric variables measured, the sum of three folds and waist girth were significantly higher as compared to the males with poor AMD. These results are in line with previous research, which showed that adolescent males and females with better AMD had a higher body fat than those with worse AMD, although the differences were not statistically significant in this case [13]. These results could be explained by the fact that, although the adolescents have an optimal AMD, the level of physical activity in the overweight group is very low (≤2.75). Thus, most of the adolescents in this group are considered physically inactive, which would make it difficult to achieve the caloric deficit necessary to reduce body fat [41]. However, the results obtained should be taken with caution, as the sample size of the overweight groups of males and females was very small, which makes it difficult to extrapolate the results. It should also be noted that, together with the results obtained in the group of overweight males, the absence of significant differences in the females and in the group of normal weight males is relevant. This could indicate that AMD alone is not so important in producing modifications in the kinanthropometric variables of adolescents, which would grant greater relevance to other elements of the diet (e.g., quantity or caloric deficit) and to other healthy lifestyle habits, such as the practice of physical activity at this age. Nevertheless, this should be confirmed in future research in which the contribution of dietary variables and physical activity to the kinanthropometric variables of adolescents is analyzed.

Regarding the second research hypothesis (H2), the results obtained allow us to partially accept it since the differences when considering BMI and AMD were significant in adolescents with optimal AMD compared to those with worse AMD. Thus, the level of physical activity practiced in males and females of the normal weight group with optimal AMD was higher. The level of physical activity and CMJ performance was higher in the males of the underweight group with optimal AMD. However, in the overweight group, the differences were significant in the kinanthropometric variables, with the males with optimal AMD showing greater body fat. Furthermore, in the females, there were only differences in the level of physical activity in the normal weight group, so future research is needed to explore the “fat but healthy diet” paradigm.

It should be noted that the findings of the present research regarding differences in physical fitness could be also influenced by the biological maturation of adolescents [42]. Adolescence is a stage in which physical and anthropometric changes occur that are determinant in the development of physical capacities. Thus, previous research has found that adolescents who mature earlier present better performance in physical condition tests, independently of the physical activity performed [42,43]. Therefore, it would be necessary that future research studying the differences in the level of physical activity practiced, kinanthropometric variables, and physical fitness variables of adolescents according to their AMD, BMI, and gender also analyze the effect of biological maturation on the changes found.

The present investigation is not free of limitations. The sample was selected by convenience in the educational centers to which we had access. It should be noted that this is a cross-sectional study in which the data were measured at a single point in time. In addition, the use of questionnaires to assess the AMD and the level of physical activity always involves a risk that adolescents will not complete the questionnaire with complete accuracy, so this is a factor to be highlighted. The classification provided by the KIDMED questionnaire makes it possible to obtain a score on the AMD, but it has gaps in terms of knowing the food intake and the quantities ingested by adolescents. Changes in physical fitness, mainly in strength, power, and cardiorespiratory fitness, could be influenced by the biological maturation of adolescents. Finally, when analyzing the results according to BMI, AMD, and gender of the adolescents, the sample sizes of some groups were too small.

Regarding the practical applications derived from the present investigation, although AMD does not seem to exert great influence on kinanthropometric variables and physical fitness in adolescents, it does seem to be related to the adoption of other healthy lifestyle habits in adolescent males and females, including a higher level of physical activity. However, the novelty of the present article with respect to previous scientific literature is that it shows the need to consider gender and BMI in the study of AMD since needs change between groups. The results obtained show that in overweight males, the optimal AMD seemed not to be so relevant for the practice of physical activity and kinanthropometric variables, and other healthy habits may be more determinant in this population, but future research is required to corroborate this. Furthermore, AMD does not seem to be a relevant variable in the improvement of the physical condition and body composition of females since the only differences were found in the group of normal weight females in the level of physical activity, but not in body composition or physical condition, independently of weight status.

The second major novelty of the present study is that the “fat but healthy diet” paradigm cannot be confirmed since differences in the level of physical activity, anthropometric variables, and physical condition according to AMD were observed only in the underweight and normal weight males. In addition, the results showed that in the overweight group, males with optimal AMD had worse body composition, and no differences were found in kinanthropometrics and physical fitness variables in any weight status group in females with different AMD. Therefore, future research is needed to determine whether AMD is a sufficient determining factor to compensate for inadequate weight status. In this sense, more scientific literature is needed to determine whether adolescents with better AMD show better body composition and physical condition, independently of their weight status, as occurs in the fat but fit paradigm [15]. Furthermore, future studies should also consider monitoring the degree of AMD during adolescence with a longitudinal and observational design mainly from pre-adolescence. This is because adolescence is a stage in which changes occur in the factors that are most determinant for the acquisition of healthy behaviors, with the influence exerted by peers increasing considerably and lessening the influence of their parents and teachers [44]; moreover, they obtain more information from other sources such as the internet or social networks [45,46], which is not always correct and can influence AMD.

## 5. Conclusions

Based on the results obtained, it can be concluded that greater AMD does not seem to produce beneficial effects in the adolescent population. Thus, only the level of physical activity showed significant differences as a function of the adolescents’ AMD, while there were no significant differences in the kinanthropometrics variables and the physical condition variables of the adolescents according to their AMD when considering the whole sample. Considering the gender of the adolescents, it was observed that the males with better AMD had a higher level of physical activity and a greater muscle mass, but also showed a greater fat mass, body mass, and BMI. As for the females with better AMD, they presented only a higher level of physical activity and higher VO_2_ max. In addition, when considering BMI and gender together, the view of the “fat but healthy diet” paradigm could not be confirmed. This is because overweight and obese males with optimal AMD showed greater body mass, sum of three skinfolds, and waist circumference, and they did not practice more physical activity than overweight males with worse AMD. In addition, no differences were found in the kinanthropometric and physical condition variables in the females in any of the weight status groups. Therefore, the fat but healthy diet paradigm cannot be confirmed in the present research.

## Figures and Tables

**Figure 1 nutrients-15-01152-f001:**
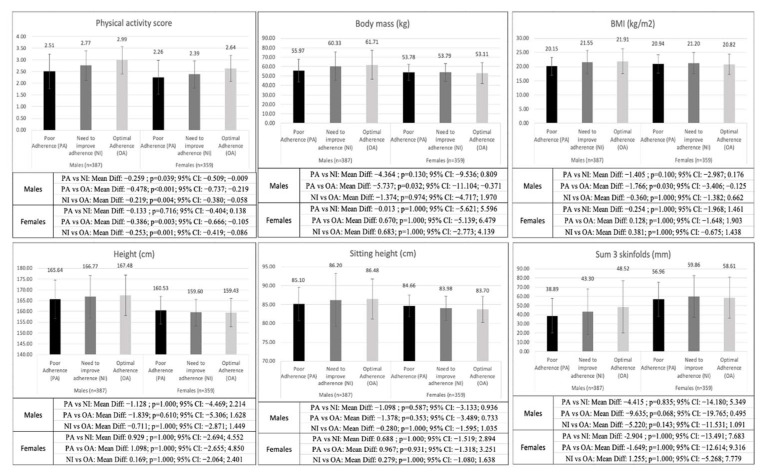
Differences in physical activity, kinanthropometric measurements, and physical fitness variables between males with different AMD and females with different AMD.

**Figure 2 nutrients-15-01152-f002:**
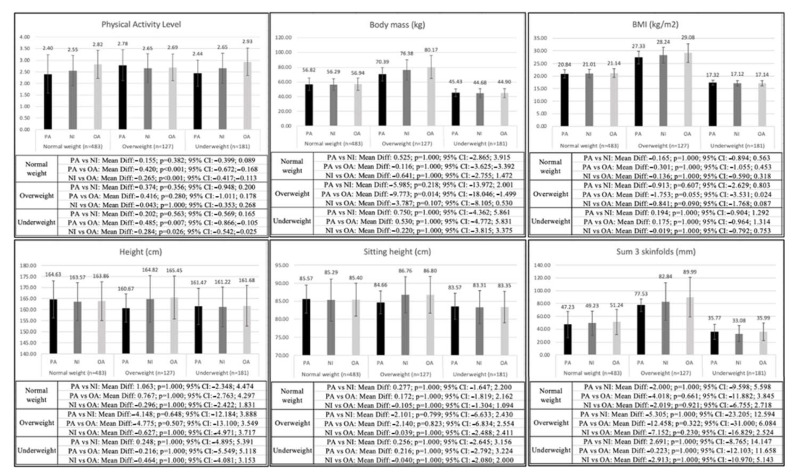
Differences in physical activity, kinanthropometric measurements, and physical fitness variables between adolescents with different AMD and weight status.

**Table 1 nutrients-15-01152-t001:** Differences in physical activity, kinanthropometric, and physical fitness variables among adolescents with different levels of AMD.

Variable	Descriptors (M ± SD)			Level of AMD			Level of AMD × Gender			Level of AMD × BMI		
	Poor Adherence (*n* = 92)	Need to Improve (*n* = 412)	Optimal Adherence (*n* = 287)	F	*p*	Effect Size (η2)	F	*p*	Effect Size (η2)	F	*p*	Effect Size (η2)
Physical activity score	2.40 ± 0.74	2.59 ± 0.64	2.82 ± 0.60	18.059	<0.001	0.046	20.271	<0.001	0.120	5.252	<0.001	0.054
Body mass (kg)	54.97 ± 10.54	57.15 ± 13.07	57.57 ± 14.00	1.173	0.310	0.003	12.062	<0.001	0.075	113.056	<0.001	0.551
BMI (kg/m^2^)	20.51 ± 3.23	21.38 ± 3.89	21.38 ± 4.08	1.729	0.178	0.005	2.098	0.064	0.014	-	-	-
Height (cm)	163.32 ± 8.28	163.28 ± 9.04	163.60 ± 9.09	0.106	0.899	0.001	29.660	<0.001	0.167	1.763	0.081	0.019
Sitting height (cm)	84.90 ± 3.81	85.12 ± 5.60	85.14 ± 4.71	0.068	0.935	0.001	8.174	<0.001	0.052	4.095	<0.001	0.043
Sum three skinfolds (mm)	47.10 ± 20.83	51.35 ± 25.13	53.38 ± 26.06	1.942	0.144	0.005	14.833	<0.001	0.091	55.955	<0.001	0.378
Waist girth (cm)	68.90 ± 6.92	69.78 ± 8.87	69.81 ± 9.15	0.354	0.702	0.001	21.064	<0.001	0.125	129.482	<0.001	0.584
Hip girth (cm)	89.18 ± 8.14	90.76 ± 9.26	91.18 ± 9.23	1.436	0.238	0.004	1.750	0.121	0.012	142.728	<0.001	0.608
Corrected arm girth (cm)	21.10 ± 2.39	21.40 ± 2.98	21.28 ± 3.19	0.347	0.707	0.001	33.501	<0.001	0.185	42.107	<0.001	0.314
Corrected thigh girth (cm)	39.34 ± 4.11	40.27 ± 4.79	40.33 ± 5.34	1.292	0.275	0.003	25.120	<0.001	0.145	40.746	<0.001	0.307
Corrected calf girth (cm)	29.07 ± 2.53	29.43 ± 3.20	29.33 ± 3.50	0.423	0.655	0.001	19.266	<0.001	0.115	15.161	<0.001	0.141
Fat mass (%)	20.86 ± 8.35	22.64 ± 10.23	23.60 ± 10.80	2.248	0.106	0.006	11.806	<0.001	0.074	60.219	<0.001	0.395
Muscle mass (kg)	18.40 ± 4.12	19.02 ± 5.05	19.04 ± 5.47	0.523	0.593	0.001	99.578	<0.001	0.402	25.097	<0.001	0.214
Waist–hip ratio	0.77 ± 0.47	0.77 ± 0.60	0.77 ± 0.52	0.820	0.441	0.002	91.963	<0.001	0.383	8.135	<0.001	0.081
VO_2_ max. (ml/kg/min)	38.97 ± 5.81	39.51 ± 5.76	40.13 ± 5.64	1.626	0.197	0.004	44.175	<0.001	0.230	4.315	<0.001	0.045
Handgrip right arm (kg)	26.75 ± 6.77	27.00 ± 8.15	26.43 ± 8.90	0.379	0.684	0.001	43.401	<0.001	0.227	9.633	<0.001	0.095
Handgrip left arm (kg)	25.19 ± 6.70	25.17 ± 7.57	24.37 ± 7.78	0.971	0.379	0.003	49.777	<0.001	0.252	9.022	<0.001	0.089
CMJ (cm)	23.55 ± 7.11	23.85 ± 6.63	23.46 ± 7.46	0.276	0.759	0.001	28.616	<0.001	0.162	4.559	<0.001	0.047

BMI: body mass index; VO_2_ max: maximum oxygen consumption; CMJ: countermovement jump.

**Table 2 nutrients-15-01152-t002:** Differences in physical activity, kinanthropometric measurements, and physical fitness variables among normal weight males with different levels of AMD and normal weight females with different levels of AMD.

Normal Weight (*n* = 483)							
Variable	Gender	Poor Adherence (A)	Need to Improve (B)	Optimal Adherence (C)	Diff A-B; *p*	Diff A-C; *p*	Diff. B-C; *p*
Physical activity score	Males (*n* = 238)	2.52 ± 0.81	2.77 ± 0.63	3.02 ± 0.57	−0.246; 0.224	−0.499; *p* = 0.001	−0.253; *p* = 0.011
	Females (*n* = 245)	2.26 ± 0.85	2.35 ± 0.60	2.61 ± 0.57	−0.088; *p* = 1.000	−0.343; *p* = 0.054	−0.255; *p* = 0.009
Body mass (kg)	Males (*n* = 238)	58.69 ± 9.42	59.30 ± 8.45	60.03 ± 8.35	−0.608; *p* = 1.000	−1.341; *p* = 1.000	−0.733; *p* = 1.000
	Females (*n* = 245)	54.87 ± 6.80	53.54 ± 6.24	53.80 ± 7.18	1.330; *p* = 1.000	1.070; *p* = 1.000	−0.259; *p* = 1.000
BMI (kg/m^2^)	Males (*n* = 238)	20.64 ± 1.65	21.11 ± 1.70	21.29 ± 1.78	−0.472; *p* = 0.811	−0.651; *p* = 0.421	−0.179; *p* = 1.000
	Females (*n* = 245)	21.06 ± 1.75	20.92 ± 1.71	21.00 ± 1.74	0.142; *p* = 1.000	0.061; *p* = 1.000	−0.081; *p* = 1.000
Height (cm)	Males (*n* = 238)	167.67 ± 8.66	167.45 ± 9.17	167.84 ± 9.06	0.216; *p* = 1.000	−0.175; *p* = 1.000	−0.391; *p* = 1.000
	Females (*n* = 245)	161.46 ± 7.16	160.01 ± 6.09	159.79 ± 6.38	1.449; *p* = 1.000	1.671; *p* = 1.000	0.222; *p* = 1.000
Sitting height (cm)	Males (*n* = 238)	85.92 ± 4.66	86.49 ± 7.72	86.86 ± 5.02	−0.562; *p* = 1.000	−0.935; *p* = 1.000	−0.373; *p* = 1.000
	Females (*n* = 245)	85.20 ± 2.94	84.20 ± 3.14	83.90 ± 3.43	0.999; *p* = 1.000	1.298; *p* = 0.787	0.299; *p* = 1.000
Sum three skinfolds (mm)	Males (*n* = 238)	37.84 ± 17.91	39.70 ± 18.25	43.98 ± 18.45	−1.857; *p* = 1.000	−6.137; *p* = 0.430	−4.280; *p* = 0.286
	Females (*n* = 245)	57.02 ± 18.50	57.95 ± 14.90	58.68 ± 17.71	−0.928; *p* = 1.000	−1.663; *p* = 1.000	−0.735; *p* = 1.000
Waist girth (cm)	Males (*n* = 238)	71.43 ± 5.42	71.31 ± 4.68	71.49 ± 4.58	0.126; *p* = 1.000	−0.056; *p* = 1.000	−0.182; *p* = 1.000
	Females (*n* = 245)	67.94 ± 4.71	66.27 ± 4.52	66.19 ± 4.83	1.664; *p* = 0.435	1.744; *p* = 0.425	0.079; *p* = 1.000
Hip girth (cm)	Males (*n* = 238)	89.50 ± 6.27	89.79 ± 5.91	90.97 ± 5.44	−0.299; *p* = 1.000	−1.473; *p* = 0.794	−1.175; *p* = 0.438
	Females (*n* = 245)	92.02 ± 5.31	91.21 ± 4.99	91.29 ± 5.18	0.809; *p* = 1.000	0.720; *p* = 1.000	−0.089; *p* = 1.000
Corrected arm girth (cm)	Males (*n* = 238)	22.65 ± 1.71	22.76 ± 2.29	22.85 ± 2.85	−0.128; *p* = 1.000	−0.201; *p* = 1.000	−0.073; *p* = 1.000
	Females (*n* = 245)	20.38 ± 1.64	20.11 ± 1.65	20.21 ± 2.02	0.263; *p* = 1.000	0.164; *p* = 1.000	−0.098; *p* = 1.000
Corrected thigh girth (cm)	Males (*n* = 238)	41.53 ± 3.61	42.36 ± 3.50	43.04 ± 4.98	−0.830; *p* = 0.966	−1.507; *p* = 0.244	−0.677; *p* = 0.601
	Females (*n* = 245)	38.61 ± 2.52	38.83 ± 3.11	38.66 ± 2.86	−0.216; *p* = 1.000	−0.046; *p* = 1.000	0.170; *p* = 1.000
Corrected calf girth (cm)	Males (*n* = 238)	30.61 ± 2.18	30.94 ± 2.47	30.62 ± 2.78	−0.330; *p* = 1.000	−0.014; *p* = 1.000	0.316; *p* = 1.000
	Females (*n* = 245)	28.68 ± 2.27	28.27 ± 1.97	28.39 ± 2.21	0.410; *p* = 1.000	0.293; *p* = 1.000	−0.117; *p* = 1.000
Fat mass (%)	Males (*n* = 238)	17.28 ± 7.62	17.97 ± 7.90	20.12 ± 8.07	−0.690; *p* = 1.000	−2.844; *p* = 0.295	−2.154; *p* = 0.123
	Females (*n* = 245)	25.02 ± 6.60	25.09 ± 5.61	25.40 ± 6.61	−0.061; *p* = 1.000	−0.375; *p* = 1.000	−0.313; *p* = 1.000
Muscle mass (kg)	Males (*n* = 238)	21.89 ± 3.13	22.36 ± 3.64	22.77 ± 4.38	−0.473; *p* = 1.000	−0.880; *p* = 0.743	−0.407; *p* = 1.000
	Females (*n* = 245)	16.05 ± 2.02	15.90 ± 2.27	15.81 ± 2.25	0.147; *p* = 1.000	0.242; *p* = 1.000	0.095; *p* = 1.000
Waist–hip ratio	Males (*n* = 238)	0.80 ± 0.04	0.80 ± 0.06	0.79 ± 0.03	0.003; *p* = 1.000	0.013; *p* = 0.602	0.010; *p* = 0.319
	Females (*n* = 245)	0.74 ± 0.03	0.73 ± 0.04	0.73 ± 0.04	0.012; *p* = 0.669	0.013; *p* = 0.551	0.002; *p* = 1.000
VO_2_ max. (ml/kg/min)	Males (*n* = 238)	41.74 ± 5.62	42.74 ± 5.39	43.24 ± 5.38	−0.998; *p* = 1.000	−1.505; *p* = 0.548	−0.507; *p* = 1.000
	Females (*n* = 245)	37.20 ± 3.74	36.34 ± 4.40	37.76 ± 4.09	0.856; *p* = 1.000	−0.565; *p* = 1.000	−1.421; *p* = 0.113
Handgrip right arm (kg)	Males (*n* = 238)	31.87 ± 6.89	31.56 ± 8.32	31.20 ± 9.34	0.307; *p* = 1.000	0.667; *p* = 1.000	0.361; *p* = 1.000
	Females (*n* = 245)	24.76 ± 3.63	23.31 ± 4.80	22.45 ± 4.91	1.454; *p* = 1.000	2.307; *p* = 0.469	0.853; *p* = 1.000
Handgrip left arm (kg)	Males (*n* = 238)	30.25 ± 7.41	29.58 ± 7.72	28.83 ± 8.18	0.669; *p* = 1.000	1.420; *p* = 0.966	0.750; *p* = 1.000
	Females (*n* = 245)	23.06 ± 3.83	21.60 ± 4.36	20.46 ± 3.84	1.463; *p* = 0.895	2.600; *p* = 0.226	1.137; *p* = 0.570
CMJ (cm)	Males (*n* = 238)	29.53 ± 7.89	27.01 ± 7.15	27.97 ± 8.10	2.515; *p* = 0.209	1.560; *p* = 0.823	−0.955; *p* = 0.823
	Females (*n* = 245)	21.12 ± 3.56	21.46 ± 5.09	21.00 ± 5.18	−0.339; *p* = 1.000	0.115; *p* = 1.000	0.454; *p* = 1.000

BMI: body mass index; VO_2_ max: maximum oxygen consumption; CMJ: countermovement jump.

**Table 3 nutrients-15-01152-t003:** Differences in physical activity, kinanthropometric measurements, and physical fitness variables among overweight males with different levels of AMD and overweight females with different levels of AMD.

Overweight (*n* = 127)							
Variable	Gender	Poor Adherence (A)	Need to Improve (B)	Optimal Adherence (C)	Diff A-B; *p*	Diff A-C; *p*	Diff. B-C; *p*
Physical activity score	Males (*n* = 76)	2.56 ± 0.80	2.76 ± 0.61	2.73 ± 0.58	−0.196; *p* = 1.000	−0.166; *p* = 1.000	0.030; *p* = 1.000
	Females (*n* = 51)	2.00 ± 0.40	2.48 ± 0.61	2.64 ± 0.57	−0.488; *p* = 0.422	−0.642; *p* = 0.198	−0.154; *p* = 1.000
Body mass (kg)	Males (*n* = 76)	74.12 ± 7.07	80.35 ± 14.56	83.72 ± 17.85	−6.227; *p* = 0.465	−9.595; *p* = 0.099	−3.369; *p* = 0.341
	Females (*n* = 51)	65.87 ± 8.91	70.33 ± 8.35	73.92 ± 6.56	−4.455; *p* = 0.965	−8.047; *p* = 0.268	−3.592; *p* = 0.592
BMI (kg/m^2^)	Males (*n* = 76)	27.21 ± 2.15	28.06 ± 2.91	29.34 ± 4.24	−0.850; *p* = 1.000	−2.129; *p* = 0.117	−1.279; *p* = 0.027
	Females (*n* = 51)	27.44 ± 3.13	28.52 ± 3.36	28.61 ± 2.26	−1.080; *p* = 0.883	−1.170; *p* = 0.843	−0.089; *p* = 1.000
Height (cm)	Males (*n* = 76)	166.20 ± 3.35	169.01 ± 10.83	168.05 ± 11.18	−2.812; *p* = 1.000	−1.855; *p* = 1.000	−0.958; *p* = 1.000
	Females (*n* = 51)	155.15 ± 1.76	158.12 ± 5.48	160.80 ± 3.48	−2.970; *p* = 1.000	−5.650; *p* = 0.666	−2.680; *p* = 0.975
Sitting height (cm)	Males (*n* = 76)	86.35 ± 0.45	88.26 ± 5.51	87.64 ± 6.06	−1.911; *p* = 1.000	−1.288; *p* = 1.000	0.623; *p* = 1.000
	Females (*n* = 51)	82.97 ± 4.00	84.37 ± 2.75	85.31 ± 2.23	−1.393; *p* = 1.000	−2.336; *p* = 1.000	−0.943; *p* = 1.000
Sum three skinfolds (mm)	Males (*n* = 76)	72.97 ± 7.15	74.52 ± 26.80	85.85 ± 35.75	−1.542; *p* = 1.000	−12.872; *p* = 0.566	−11.330; *p* = 0.044
	Females (*n* = 51)	82.09 ± 10.80	96.15 ± 29.36	97.38 ± 19.19	−14.061; *p* = 0.453	−15.298; *p* = 0.413	−1.237; *p* = 1.000
Waist girth (cm)	Males (*n* = 76)	81.44 ± 4.21	86.97 ± 7.59	89.02 ± 7.98	−5.535; *p* = 0.110	−7.584; *p* = 0.016	−2.049; *p* = 0.334
	Females (*n* = 51)	79.76 ± 5.22	79.14 ± 8.33	79.42 ± 7.86	0.627; *p* = 1.000	0.341; *p* = 1.000	−0.286; *p* = 1.000
Hip girth (cm)	Males (*n* = 76)	102.95 ± 5.02	104.54 ± 8.27	106.49 ± 9.71	−1.589; *p* = 1.000	−3.541; *p* = 0.753	−1.952; *p* = 0.544
	Females (*n* = 51)	101.67 ± 4.00	104.22 ± 7.05	105.62 ± 4.44	−2.547; *p* = 1.000	−3.950; *p* = 0.671	−1.403; *p* = 1.000
Corrected arm girth (cm)	Males (*n* = 76)	25.52 ± 2.38	25.49 ± 2.88	25.36 ± 3.42	0.027; *p* = 1.000	0.156; *p* = 1.000	0.129; *p* = 1.000
	Females (*n* = 51)	22.59 ± 1.86	23.38 ± 3.03	21.74 ± 2.22	−0.782; *p* = 1.000	0.853; *p* = 1.000	1.635; *p* = 0.077
Corrected thigh girth (cm)	Males (*n* = 76)	45.66 ± 3.60	46.57 ± 5.40	45.90 ± 6.27	−0.910; *p* = 1.000	−0.236; *p* = 1.000	0.674; *p* = 1.000
	Females (*n* = 51)	43.03 ± 4.38	41.57 ± 5.26	43.98 ± 3.83	1.463; *p* = 1.000	−0.954; *p* = 1.000	−2.418; *p* = 0.161
Corrected calf girth (cm)	Males (*n* = 76)	31.43 ± 3.03	31.94 ± 3.72	31.85 ± 3.75	−0.509; *p* = 1.000	−0.412; *p* = 1.000	0.097; *p* = 1.000
	Females (*n* = 51)	29.70 ± 2.28	30.59 ± 6.20	31.06 ± 8.43	−0.886; *p* = 1.000	−1.360; *p* = 1.000	−0.474; *p* = 1.000
Fat mass (%)	Males (*n* = 76)	31.01 ± 4.24	34.12 ± 11.64	38.05 ± 15.80	−3.118; *p* = 1.000	−7.049; *p* = 0.239	−3.931; *p* = 0.117
	Females (*n* = 51)	34.36 ± 5.36	38.64 ± 11.00	40.83 ± 6.93	−4.280; *p* = 0.861	−6.477; *p* = 0.378	−2.197; *p* = 1.000
Muscle mass (kg)	Males (*n* = 76)	25.08 ± 2.97	26.31 ± 5.12	25.90 ± 6.33	−1.234; *p* = 1.000	−0.817; *p* = 1.000	0.416; *p* = 1.000
	Females (*n* = 51)	18.53 ± 2.73	18.82 ± 3.37	19.51 ± 3.38	−0.283; *p* = 1.000	−0.973; *p* = 1.000	−0.690; *p* = 1.000
Waist–hip ratio	Males (*n* = 76)	0.79 ± 0.02	0.83 ± 0.04	0.84 ± 0.05	−0.042; *p* = 0.185	−0.046; *p* = 0.129	−0.005; *p* = 1.000
	Females (*n* = 51)	0.78 ± 0.03	0.76 ± 0.05	0.75 ± 0.06	0.025; *p* = 0.808	0.033; *p* = 0.509	0.008; *p* = 1.000
VO_2_ max. (ml/kg/min)	Males (*n* = 76)	37.51 ± 5.89	40.32 ± 4.99	39.04 ± 5.19	−2.812; *p* = 0.820	−1.529; *p* = 1.000	1.283; *p* = 0.912
	Females (*n* = 51)	32.43 ± 5.34	33.88 ± 3.59	33.52 ± 2.89	−1.448; *p* = 1.000	−1.092; *p* = 1.000	0.356; *p* = 1.000
Handgrip right arm (kg)	Males (*n* = 76)	35.13 ± 3.52	34.87 ± 8.38	33.54 ± 12.82	0.255; *p* = 1.000	1.581; *p* = 1.000	1.326; *p* = 1.000
	Females (*n* = 51)	24.15 ± 4.13	25.65 ± 5.77	24.56 ± 2.97	−1.502; *p* = 1.000	−0.414; *p* = 1.000	1.088; *p* = 1.000
Handgrip left arm (kg)	Males (*n* = 76)	29.98 ± 6.32	32.70 ± 7.85	30.67 ± 10.64	−2.728; *p* = 1.000	−0.697; *p* = 1.000	2.031; *p* = 0.601
	Females (*n* = 51)	23.73 ± 4.46	23.06 ± 4.60	22.87 ± 3.21	0.669; *p* = 1.000	0.854; *p* = 1.000	0.185; *p* = 1.000
CMJ (cm)	Males (*n* = 76)	20.93 ± 4.30	24.13 ± 5.41	20.55 ± 9.11	−3.208; *p* = 0.969	0.377; *p* = 1.000	3.585; *p* = 0.070
	Females (*n* = 51)	20.25 ± 2.70	18.54 ± 5.95	16.15 ± 3.66	1.710; *p* = 1.000	4.100; *p* = 0.728	2.390; *p* = 0.743

BMI: body mass index; VO_2_ max: maximum oxygen consumption; CMJ: countermovement jump.

**Table 4 nutrients-15-01152-t004:** Differences in physical activity, kinanthropometric measurements, and physical fitness variables among underweight males with different levels of AMD and underweight females with different levels of AMD.

Underweight (*n* = 181)							
Variable	Gender	Poor Adherence (A)	Need to Improve (B)	Optimal Adherence (C)	Diff A-B; *p*	Diff A-C; *p*	Diff. B-C; *p*
Physical activity score	Males (*n* = 95)	2.48 ± 0.65	2.79 ± 0.69	3.13 ± 0.60	−0.313; *p* = 0.291	−0.646; *p* = 0.004	−0.333; *p* = 0.070
	Females (*n* = 86)	2.38 ± 0.42	2.47 ± 0.54	2.75 ± 0.53	−0.092; *p* = 1.000	−0.372; *p* = 0.378	−0.280; *p* = 0.174
Body mass (kg)	Males (*n* = 95)	45.90 ± 6.45	45.23 ± 7.11	47.85 ± 6.24	0.674; *p* = 1.000	−1.952; *p* = 1.000	−2.625; *p* = 0.560
	Females (*n* = 86)	44.60 ± 1.96	44.03 ± 4.41	42.23 ± 4.63	0.573; *p* = 1.000	2.372; *p* = 1.000	1.799; *p* = 1.000
BMI (kg/m^2^)	Males (*n* = 95)	17.29 ± 1.12	16.93 ± 1.03	17.38 ± 0.94	0.352; *p* = 1.000	−0.092; *p* = 1.000	−0.444; *p* = 0.989
	Females (*n* = 86)	17.37 ± 0.52	17.34 ± 0.87	16.93 ± 1.02	0.024; *p* = 1.000	0.442; *p* = 1.000	0.418; *p* = 1.000
Height (cm)	Males (*n* = 95)	162.01 ± 9.94	162.97 ± 10.38	165.89 ± 9.07	−0.961; *p* = 1.000	−3.888; *p* = 0.430	−2.927; *p* = 0.396
	Females (*n* = 86)	160.52 ± 3.99	159.15 ± 6.49	157.87 ± 7.71	1.375; *p* = 1.000	2.657; *p* = 1.000	1.282; *p* = 1.000
Sitting height (cm)	Males (*n* = 95)	83.34 ± 4.41	83.61 ± 5.23	84.34 ± 5.18	−0.275; *p* = 1.000	−1.007; *p* = 1.000	−0.732; *p* = 1.000
	Females (*n* = 86)	83.97 ± 1.82	82.96 ± 3.81	82.45 ± 3.42	1.016; *p* = 1.000	1.520; *p* = 1.000	0.505; *p* = 1.000
Sum three skinfolds (mm)	Males (*n* = 95)	30.94 ± 11.02	25.09 ± 6.37	29.98 ± 14.17	5.851; *p* = 0.878	0.960; *p* = 1.000	−4.891; *p* = 0.775
	Females (*n* = 86)	44.21 ± 7.35	42.53 ± 10.78	41.43 ± 10.36	1.677; *p* = 1.000	2.776; *p* = 1.000	1.098; *p* = 1.000
Waist girth (cm)	Males (*n* = 95)	64.10 ± 3.56	63.36 ± 3.88	65.28 ± 5.10	0.744; *p* = 1.000	−1.182; *p* = 1.000	−1.926; *p* = 0.327
	Females (*n* = 86)	60.77 ± 2.77	60.35 ± 2.50	59.71 ± 2.13	0.429; *p* = 1.000	1.061; *p* = 1.000	0.632; *p* = 1.000
Hip girth (cm)	Males (*n* = 95)	80.02 ± 4.89	78.97 ± 5.27	82.34 ± 5.47	1.051; *p* = 1.000	−2.325; *p* = 0.637	−3.376; *p* = 0.040
	Females (*n* = 86)	82.99 ± 2.14	82.86 ± 4.13	81.17 ± 4.59	0.135; *p* = 1.000	1.817; *p* = 1.000	1.683; *p* = 0.662
Corrected arm girth (cm)	Males (*n* = 95)	19.64 ± 1.08	19.52 ± 1.97	19.87 ± 2.24	0.113; *p* = 1.000	−0.236; *p* = 1.000	−0.349; *p* = 1.000
	Females (*n* = 86)	18.19 ± 1.26	18.06 ± 1.20	17.73 ± 1.25	0.124; *p* = 1.000	0.454; *p* = 1.000	0.331; *p* = 1.000
Corrected thigh girth (cm)	Males (*n* = 95)	36.87 ± 2.50	37.07 ± 3.17	37.18 ± 3.17	−0.200; *p* = 1.000	−0.306; *p* = 1.000	−0.105; *p* = 1.000
	Females (*n* = 86)	34.18 ± 2.03	34.90 ± 2.26	34.31 ± 2.52	−0.721; *p* = 1.000	−0.125; *p* = 1.000	0.596; *p* = 1.000
Corrected calf girth (cm)	Males (*n* = 95)	27.91 ± 1.70	28.25 ± 2.09	28.44 ± 2.43	−0.347; *p* = 1.000	−0.535; *p* = 1.000	−0.188; *p* = 1.000
	Females (*n* = 86)	26.14 ± 1.34	26.72 ± 1.59	26.36 ± 1.63	−0.576; *p* = 1.000	−0.220; *p* = 1.000	0.355; *p* = 1.000
Fat mass (%)	Males (*n* = 95)	14.05 ± 4.86	11.55 ± 2.69	13.80 ± 5.88	2.495; *p* = 0.824	0.247; *p* = 1.000	−2.249; *p* = 0.617
	Females (*n* = 86)	19.70 ± 2.52	19.39 ± 3.81	18.30 ± 3.54	0.311; *p* = 1.000	1.405; *p* = 1.000	1.094; *p* = 1.000
Muscle mass (kg)	Males (*n* = 95)	17.45 ± 2.21	17.75 ± 2.95	18.22 ± 2.94	−0.296; *p* = 1.000	−0.773; *p* = 1.000	−0.477; *p* = 1.000
	Females (*n* = 86)	12.92 ± 1.31	13.15 ± 1.41	12.67 ± 1.56	−0.233; *p* = 1.000	0.249; *p* = 1.000	0.482; *p* = 1.000
Waist–hip ratio	Males (*n* = 95)	0.80 ± 0.04	0.80 ± 0.04	0.79 ± 0.04	−0.002; *p* = 1.000	0.009; *p* = 1.000	0.011; *p* = 0.882
	Females (*n* = 86)	0.73 ± 0.05	0.73 ± 0.02	0.74 ± 0.04	0.004; *p* = 1.000	−0.004; *p* = 1.000	−0.008; *p* = 1.000
VO_2_ max. (ml/kg/min)	Males (*n* = 95)	40.81 ± 7.40	43.76 ± 5.14	42.33 ± 6.28	−2.942; *p* = 0.149	−1.513; *p* = 1.000	1.429; *p* = 0.661
	Females (*n* = 86)	36.51 ± 3.68	38.05 ± 3.69	39.72 ± 4.96	−1.543; *p* = 1.000	−3.219; *p* = 0.289	−1.676; *p* = 0.461
Handgrip right arm (kg)	Males (*n* = 95)	22.15 ± 5.70	25.07 ± 7.36	26.34 ± 6.01	−2.919; *p* = 0.505	−4.188; *p* = 0.190	−1.269; *p* = 1.000
	Females (*n* = 86)	22.30 ± 3.88	20.20 ± 3.86	19.33 ± 4.51	2.100; *p* = 1.000	2.966; *p* = 0.836	0.866; *p* = 1.000
Handgrip left arm (kg)	Males (*n* = 95)	21.63 ± 5.64	23.51 ± 6.51	24.57 ± 5.71	−1.880; *p* = 0.969	−2.937; *p* = 0.440	−1.057; *p* = 1.000
	Females (*n* = 86)	20.66 ± 3.14	19.01 ± 3.57	18.19 ± 3.44	1.657; *p* = 1.000	2.472; *p* = 0.944	0.815; *p* = 1.000
CMJ (cm)	Males (*n* = 95)	21.55 ± 7.15	27.76 ± 6.15	25.43 ± 5.20	−6.208; *p* = 0.003	−3.881; *p* = 0.163	2.327; *p* = 0.344
	Females (*n* = 86)	19.05 ± 3.84	20.75 ± 4.27	21.39 ± 4.98	−1.699; *p* = 1.000	−2.344; *p* = 1.000	−0.644; *p* = 1.000

BMI: body mass index; VO_2_ max: maximum oxygen consumption; CMJ: countermovement jump.

## Data Availability

The data presented in this study are available on request from the corresponding author. The data are not publicly available because they contain information that could compromise the privacy of research participants, but they are available from the corresponding author upon reasonable request.

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
