# Peer review of "Differences in Kinanthropometric Variables and Physical Fitness of Adolescents with Different Adherence to the Mediterranean Diet and Weight Status: “Fat but Healthy Diet” Paradigm"

_nutrients, 2023, doi:10.3390/nu15051152_

Round 1

Reviewer 1 Report

This an interesting topic, trying to associate dietary practices with other parameters of fitness, endurance and strength.  The over all presentation is good, since the breakdown allows a reader to entertain other possible ideas that may be hidden in the data.  Is any follow up being considered? It would be interesting to observe any changes in the classifications of subjects 1 or 2 years later, at least the younger ones, as well as reasons for this.  Also, can you comment on whether the schools in question have courses on nutrition as electives or integrated into mandatory subjects, perhaps as part of PE?  As always in such studies, there is reliance on self-assessment, which seems to be addressed as much as possible.   

Author Response

Reviewer 1

This an interesting topic, trying to associate dietary practices with other parameters of fitness, endurance and strength.  The overall presentation is good, since the breakdown allows a reader to entertain other possible ideas that may be hidden in the data. 

+ Dear reviewer, thank you very much for reviewing our manuscript. We have made all the considerations you indicated to increase the quality of the manuscript. We hope that the final result will be satisfactory for you.

Is any follow up being considered? It would be interesting to observe any changes in the classifications of subjects 1 or 2 years later, at least the younger ones, as well as reasons for this. 

+ Thank you very much for your contribution. A new line of future research has been included in which to consider progression in AMD in younger adolescents, as the factors influencing them are continually changing during this stage of their lives.

Also, can you comment on whether the schools in question have courses on nutrition as electives or integrated into mandatory subjects, perhaps as part of PE?  As always in such studies, there is reliance on self-assessment, which seems to be addressed as much as possible.

+ Thank you very much for your comment. We have included in the method those subjects of compulsory secondary education that in Spain can provide information on AMD, but always in a secondary way.

Reviewer 2 Report

The authors have presented a very well-executed and relevant study with 791 subjects. The study reports interesting data about adherence to the Mediterranean diet with regard to physical activity and kinanthropometric variables. The results show that there is no benefit of the Mediterranean diet on body composition (fat mass, muscle mass, BMI) or strength (VO2 max, grip strength or CMJ), still the authors have concluded that better AMD is associated with better kinanthropometric variables. The authors should kindly not ignore the results and rewrite the manuscript in order to reflect the results reported.

 Specific comments:

1)    Please add more details about “Fat but healthy diet” paradigm for better understanding of the readers.

2)    Line 310: the difference in fat mass is much bigger (20.86% vs 23.60%) than muscle mass (18.4% vs 19.04%), yet the authors have drawn the conclusion that higher BMI is coming from increased muscle mass. If addition muscle was the source, why don't we see any benefit in measures of strength (grip strength or CMJ)? Please moderate as well as discuss the presence of a higher fat mass in AMD groups.

3)    The only other significant difference in the level of physical activity, which could very well be awareness driven whereby adolescents with more awareness about “healthy dietary pattern” also participates in more physical activity. However, the results do not show any benefits of the Mediterranean diet on health parameters including BMI, % fat mass, % muscle mass, VO2 max, hand grip or CMJ. The marginally (non-significant) better VO2 max could be the result of higher physical activity.

4)    The lack of benefits of the Mediterranean diet should be explicitly pointed out in the abstract as well as the conclusion of the manuscript to reflect the results reported in the manuscript.

5)    Please avoid very long sentences for easier reading.

Author Response

Reviewer 2

The authors have presented a very well-executed and relevant study with 791 subjects. The study reports interesting data about adherence to the Mediterranean diet with regard to physical activity and kinanthropometric variables.

+ Dear reviewer, thank you very much for reviewing our manuscript and providing input for its improvement. We have tried to comply with all of them to increase the quality of the manuscript. We believe that the current version is much better than the previous one. We look forward to hearing from you if there are any further improvements to be made.

The results show that there is no benefit of the Mediterranean diet on body composition (fat mass, muscle mass, BMI) or strength (VO2 max, grip strength or CMJ), still the authors have concluded that better AMD is associated with better kinanthropometric variables. The authors should kindly not ignore the results and rewrite the manuscript in order to reflect the results reported.

+ Thank you very much for your contribution. It has been specified in the conclusions section that no significant differences were found in body composition and fitness of adolescents according to their AMD.

Specific comments:

1) Please add more details about “Fat but healthy diet” paradigm for better understanding of the readers.

+ Thank you for your great contribution. We have included information on the ”fat but healthy diet paradigm” in the introduction and discussion for a better understanding of the paradigm.

2) Line 310: the difference in fat mass is much bigger (20.86% vs 23.60%) than muscle mass (18.4% vs 19.04%), yet the authors have drawn the conclusion that higher BMI is coming from increased muscle mass. If addition muscle was the source, why don't we see any benefit in measures of strength (grip strength or CMJ)? Please moderate as well as discuss the presence of a higher fat mass in AMD groups.

+ Dear reviewer, you are correct in your proposal. We have modified the discussion as well as the conclusions in this regard.

3) The only other significant difference in the level of physical activity, which could very well be awareness driven whereby adolescents with more awareness about “healthy dietary pattern” also participates in more physical activity. However, the results do not show any benefits of the Mediterranean diet on health parameters including BMI, % fat mass, % muscle mass, VO2 max, hand grip or CMJ. The marginally (non-significant) better VO2 max could be the result of higher physical activity.

+ Thank you very much for your contribution. The discussion and conclusion have been modified to include what you indicate, and it seems to us to be an accurate view of the results.

4) The lack of benefits of the Mediterranean diet should be explicitly pointed out in the abstract as well as the conclusion of the manuscript to reflect the results reported in the manuscript.

+ Thank you very much for your contribution. The abstract and conclusions have been modified to clarify the results obtained.

5) Please avoid very long sentences for easier reading.

+ Thank you very much. The entire manuscript has been revised by rewriting the sentences that were excessively long.

Round 2

Reviewer 2 Report

The authors have made all the necessary changes to address my questions.